# Survey of Pharmacists’ Knowledge, Attitudes, and Practices (KAP) concerning COVID-19 Infection Control after Being Involved in Vaccine Preparation: A Cross-Sectional Study

**DOI:** 10.3390/ijerph19159035

**Published:** 2022-07-25

**Authors:** Nobuyuki Wakui, Mayumi Kikuchi, Risa Ebizuka, Takahiro Yanagiya, Chikako Togawa, Raini Matsuoka, Nobutomo Ikarashi, Miho Yamamura, Shunsuke Shirozu, Yoshiaki Machida, Kenichi Suzuki, Hajime Kato

**Affiliations:** 1Division of Applied Pharmaceutical Education and Research, Hoshi University, 2-4-41 Ebara, Shinagawa-ku, Tokyo 142-8501, Japan; s171038@hoshi.ac.jp (R.E.); s181156@hoshi.ac.jp (C.T.); s181219@hoshi.ac.jp (R.M.); m-yamamura@hoshi.ac.jp (M.Y.); s-shirozu@hoshi.ac.jp (S.S.); y-machida@hoshi.ac.jp (Y.M.); kenichi-suzuki@hoshi.ac.jp (K.S.); 2Shinagawa Pharmaceutical Association, 2-4-2 Nakanobu, Shinagawa-ku, Tokyo 142-0053, Japan; tomato_5mk@yahoo.co.jp (M.K.); yanagiya@tnb.co.jp (T.Y.); sina89@jewel.ocn.ne.jp (H.K.); 3Department of Biomolecular Pharmacology, Hoshi University, 2-4-41 Ebara, Shinagawa-ku, Tokyo 142-8501, Japan; ikarashi@hoshi.ac.jp

**Keywords:** vaccinations, pharmacist, COVID-19, KAP, public health, infection control

## Abstract

Vaccination is crucial for preventing the spread of COVID-19. Vaccination for COVID-19 was implemented in Japan in community units, and community pharmacists were engaged in vaccine preparation. Capturing the knowledge, attitudes, and practices (KAP) of pharmacists regarding COVID-19 infection control is important for developing future community health action strategies and plans. We conducted a cross-sectional study among 141 pharmacists who were members of a pharmacist association in the Shinagawa Ward of Tokyo (1–31 July 2021) using a Google online questionnaire. The questionnaire included demographic information and KAP questions regarding COVID-19. A correlation test was used for analyzing KAP scores. Significant correlations were found among all KAP scores. Stepwise logistic regression analysis showed “age” as a significant knowledge factor and “marriage”, “pharmacist careers”, “information source: official government website”, and “information source: word of mouth from family and friends” as significant attitude factors. Good KAP scores were recorded in this study, indicating increased comprehension of infection control measures and increased knowledge scores, as pharmacy pharmacists were practically involved in COVID-19 infection control measures through vaccine preparation. Policymakers should understand the value of pharmacists as healthcare professionals and should enhance public health through the effective use of pharmacists.

## 1. Introduction

Severe acute respiratory syndrome caused by a novel coronavirus was first reported in China in December 2019 [1]. Later, this severe acute respiratory syndrome coronavirus 2 (SARS-CoV-2) was found to be highly contagious and resulted in life-threatening and complex respiratory illnesses [2]. The World Health Organization declared a public health emergency in January 2020, which made it an international concern [3]. Currently, the spread of SARS-CoV-2 has become a serious public health issue worldwide [4].

With the spread of SARS-CoV-2 and the unprecedented growing demand for associated healthcare, public health measures to stop the spread of infection are crucial [5]. Many people worldwide obtain information on infectious disease control by professionals through the mass media [6,7]. However, because the prevalence of COVID-19 varies widely in different regions [8,9], direct responses and guidance according to the context of each region are required. In this regard, pharmacists from all over the world are in a position to be directly or indirectly involved with patients and the local population, thus rapidly responding to public health challenges to meet local circumstances [10] and playing a key role in preventing the spread of COVID-19 [11]. Specifically, these responses include administering vaccines [12,13], educating the population regarding practices and attitudes toward preventing infection [14], participating in the supply and management of medicines [14], and actively conducting drug treatment evaluations and side effect monitoring [15]. These responses have curtailed the spread of COVID-19 and reduced the burden on medical facilities. As a result, in addition to contributing to the avoidance of healthcare disruptions [16], pharmacists provided added value to patients and the entire healthcare system and gained high confidence from the local population [17]. In Japan, pharmacists have been engaged in vaccine filling and the drug management of vaccinations for COVID-19 since June 2021. Thereby, vaccinations of local residents were efficiently carried out.

Awareness toward countermeasures is important for COVID-19 infection control, and a survey of knowledge, attitudes, and practices (KAPs) related to COVID-19 according to KAP theory [18] has been conducted on healthcare professionals in many countries [19,20,21,22]. In addition, a KAP survey for pharmacists has been reported [23,24,25,26]. However, these survey findings were conducted prior to COVID-19 vaccine development, and pharmacists’ KAP information has not been reported.

The spread of the COVID-19 infection is still ongoing, and the prospects for complete convergence remain opaque. In such a situation, pharmacists play an important role through their involvement in a vaccination program to prevent the spread of the COVID-19 infection. Awareness of the KAP status for COVID-19 infection control among pharmacists after being involved with vaccinations will be informative in determining future community health action strategies and plans from the perspective of improving and promoting community public health. Therefore, we conducted a KAP survey on infection control measures for COVID-19 among pharmacists.

## 2. Materials and Methods

### 2.1. Study Design and Setting

This was a cross-sectional study conducted with pharmacists who were members of a pharmacist association in the Shinagawa Ward of Tokyo, Japan. Data collection took place between 1 July 2021, and 31 July 2021. The survey was conducted online via a web questionnaire that used a Google form. Participants responded anonymously to questions on demographics and their KAP regarding COVID-19. Of the 255 pharmacists surveyed, there were 141 respondents (a response rate of 55.3%), and all responded to all items and were included in the analysis. All participants responded anonymously (Figure 1).

### 2.2. Content of the Survey Instrument

Data were collected from pharmacists to assess KAP on infection control measures for COVID-19. The questionnaire consisted of two parts: demographic information and KAP questions. The KAP questions assessed COVID-19 knowledge, attitudes, and practices. The knowledge section consisted of 17 questions, and participants responded to each question with a selection formula of “Yes”, “No”, and “Don’t know”. These questions assessed their level of knowledge of the clinical manifestations, transmission route, prevention, and management of COVID-19. The attitude section consisted of 16 questions that assessed attitudes regarding vaccinating against COVID-19 and preventing the spread of infection. The practice section consisted of 11 questions that assessed the measures that were taken during daily living to prevent COVID-19 infection. Regarding the items of attitudes and practices, the selection formula was “yes” or “no”. In addition, as an assessment of the internal consistency of the KAP questionnaire, a reliability analysis was performed using Cronbach’s alpha coefficient, which showed good reliability values of 0.80 for knowledge and 0.74 for attitudes, and an acceptable value of 0.53 for practices.

### 2.3. Definition of Terms

The Japanese School Education Law was revised in 2006, and pharmacy universities shifted from a 4-year system to the current 6-year system to strengthen clinical education. With this revision, only those who had been educated at a 6-year pharmacy university could take the national pharmacist examination. Therefore, some of the participants in this survey were pharmacists who graduated from a 4-year pharmacy school before the revision of the law, and some were pharmacists who graduated from a 6-year pharmacy school.

### 2.4. Study Variables

Demographic variables included age, sex (men and women), marital status (married and unmarried), academic background (4-year university, 6-year university, master’s degree, and doctoral degree), and the information source of COVID-19. Knowledge, attitude, and practice scores were labeled as adequate knowledge, positive attitudes, and good practices, respectively, for knowledge, attitude, and practice scores based on Bloom’s cutoffs and using 80% or more of the scores as the reference standard [27,28].

Knowledge scores were obtained by assigning 1 point for each correct answer, assigning 0 to an incorrect/“Don’t know” answer, and summing all items. Knowledge scores ranged from 0 to 17 points, with 14 points or more labeled as adequate knowledge regarding COVID-19 and less than 14 points as inadequate knowledge.

For attitude scores, we assigned 1 point for positive attitudes and 0 points for negative attitudes for each question item and summed those items. Attitude scores ranged from 0 to 16 points, with 13 or more labeled as a positive attitude and less than 13 as a negative attitude.

For practice scores, a score of 1 was assigned if answers reflecting good practice were selected, 0 was assigned if answers reflecting bad practice were chosen, and those items were summed. Practice scores ranged from 1 to 11 points, with more than 9 points labeled as good practice and less than 9 as bad practice.

### 2.5. Statistical Analysis

The resulting data were encoded, validated, and analyzed using the statistical software R version 4.0.2 (R Foundation for Statistical Computing, Vienna, Austria). By setting the Google form so that it could not be finished without answers, there were 0 missing data. Numerical data were summarized with the mean and standard deviation. Categorical data were summarized by frequency and ratio. A correlation test was used to determine associations among knowledge, attitude, and practice scores. The assessment of the association between demographic variables and knowledge, attitude, and practice scores was conducted using a logistic regression analysis. A *p* value of less than 0.05 was considered statistically significant.

### 2.6. Ethical Considerations

This study was reviewed and approved by the Institutional Review Board Committee of Hoshi University (Approval No. 2021-01). Signed informed consent was obtained from all respondents, and participation was voluntary.

## 3. Results

### 3.1. Participant Information

The majority of the participants were women (65.2%) with a mean age of 44.8 years (SD: 11.9). Pharmacist career scores averaged 16.1 years (SD: 10.5), and 97 participants were married (68.8%). The academic background of the 141 participants included 93 (66.0%) bachelor’s degrees in pharmacy in the fourth grade, 32 (22.7%) bachelor’s degrees in pharmacy in the sixth grade, 13 (9.2%) master’s degrees, and 3 (2.1%) doctoral degrees. The main sources of information on COVID-19 were 32 (22.7%) international health organizations, such as the CDC and WHO, 101 (71.6%) governmental office internet sites, 124 (87.9%) news media sources, 50 (35.5%) social media sources, 28 (19.9%) cases of word of mouth from family/friends, and 12 (8.5%) original papers (Table 1).

### 3.2. Pharmacists’ Knowledge of COVID-19

Table 2 shows the results regarding knowledge of COVID-19. Most respondents answered 15 of 17 knowledge items correctly (M = 14.4, SD = 2.5). Ten items had correct answer rates of over 90%. Conversely, nearly 75% of the respondents could not answer the question that included “sneezing, runny nose, stuffy nose, and headache are not common symptoms of COVID-19.” Regarding the item “COVID-19 is caused by beta coronavirus”, only 41.8% answered correctly, and nearly half (46.8%) answered that they did not know. Regarding the question that COVID-19 is transmitted by the ingestion of food, 70.9% of the respondents answered correctly, 17% answered incorrectly, and 12.1% answered that they did not know. On the basis of Bloom’s cutoff values, 118 (83.7%) of the pharmacists were labeled as having good knowledge, with a knowledge score of 14 or higher, and 23 (16.3%) pharmacists were labeled as having inadequate knowledge, with a knowledge score of less than 14.

### 3.3. Attitudes of Pharmacists regarding COVID-19

Table 3 shows the results regarding attitudes of COVID-19. Many respondents indicated positive attitudes for 13 of 16 items (M = 13.0, SD = 2.6). In particular, for six items, including “I would like to be vaccinated with COVID-19” (94.3%), “Pharmacists can contribute to the promotion of vaccination of local residents from the public health and social aspects” (91.5%), “Vaccination can prevent the spread of COVID-19” (94.3%), “all information regarding COVID-19 needs to be shared with other healthcare professionals” (96.5%), “it is the pharmacist’s social mission to work together to curb the spread of COVID-19” (97.9%), and “I understand that this infection is very infectious” (97.9%), more than 90% of the pharmacists responded with a positive attitude. Conversely, for items related to ultimately being able to fully control COVID-19 infections (31.9%), there were fewer positive attitude responses. On the basis of Bloom’s cutoff score, there were 95 (67.4%) pharmacists who were considered to have positive attitudes with an attitude score of 13 points or more, and 46 (32.6%) pharmacists who were considered to have negative attitudes with an attitude score of less than 13 points.

### 3.4. Practices of Pharmacists regarding COVID-19

Most of the respondents answered that they were performing 10 of 11 practice items (M = 9.4, SD = 1.4). Six items, including keeping a social distance (97.9%), performing handwashing with soap routinely (97.9%), avoiding crowds (91.5%), handling possessions of patients suspected of having the coronavirus with infection prevention measures (96.5%), covering the nose and mouth with tissues and handkerchiefs when sneezing or coughing (97.2%), and ensuring that used tissues were discarded in a trash box to prevent infection (97.9%), were scored at 90% or more for good practices, and the majority also indicated good practices for other items. Conversely, items such as “I tried not to take the elevator as much as possible” (41.1%) and “I have attended workshop for COVID-19 infection prevention” (53.9%) had low values (Table 4). On the basis of Bloom’s cutoff values, 111 (78.7%) of the pharmacists were labeled with good behaviors with a score of 9 or higher, and 30 (21.3%) were labeled with bad behaviors with a score of less than 9 points.

### 3.5. Correlations between Knowledge, Attitude, and Practice Scores

Knowledge, attitude, and practice scores showed significant associations with each other (Table 5). A strong correlation was found in the order of knowledge and attitude (r = 0.44, *p* < 0.001) > attitude and practice (r = 0.32, *p* < 0.001) > knowledge and practice (r = 0.21, *p* = 0.01).

### 3.6. Relationship between Participants’ Attributes and KAP Scores

Univariate logistic regression analysis was used to assess attribute factors associated with KAP (Table 6). The findings showed that chronological age (30 s, OR: 6.3, 95% CI: 1.6–25.1; 40 s, OR: 11.1, 95% CI: 2.5–48.8; 50 s, OR: 3.8, 95% CI: 1.1–13.3; and 60 s, OR: 16.2, 95% CI: 1.78–147.1), marital status (OR: 2.9, 95% CI: 1.2–7.3), academic background (6 years-university, OR: 0.3, 95% CI: 0.1–0.8), and pharmacist career (over 20 years, OR: 4.1, 95% CI: 1.2–13.9) were factors associated with knowledge. Marital status (OR: 0.42, 95% CI: 0.18–0.97) and source of information on COVID-19 (government official website and government official site, OR: 2.5, 95% CI: 1.2–5.3; and word of mouth from family/friends, OR: 0.4, 95% CI: 0.2–0.9) were associated with attitudes. No factors associated with practices were found (Table 6).

### 3.7. Extraction of Factors Associated with KAP Scores by Stepwise Logistic Regression Analysis

Factors associated with KAP were extracted using stepwise multiple logistic regression analysis with factors that had a *p* value less than 0.20 as a result of univariate logistic regression analysis (Table 7). As a result, “age” was extracted as an important variable for knowledge. Regarding attitudes, “marriage”, “academic background”, “pharmacist careers”, “information source: official government website”, and “information source: word of mouth from family and friends” were extracted as important factors. Regarding practices, no factors showing an association were extracted.

The odds ratios and confidence intervals for “age” in knowledge were as follows: ages in their 20 s (OR = 6.3, 95% CI = 1.58–25.08), 30 s (OR = 11.1, 95% CI = 2.52–48.84), 40 s (OR = 3.75, 95% CI = 1.06–13.31), 50 s (OR = 3.75, 95% CI = 1.06–13.31), and 60 s (OR = 16.2, 95% CI = 1.78–147.06).

The odds ratio and confidence interval for “marriage” in attitudes were OR = 0.3 and 95% CI = 0.10–0.94 for married people. The odds ratio and confidence interval for academic background were OR = 0.3 and 95% CI = 0.10–0.94 for graduating from pharmacy studies in a 6-year-university. Regarding information sources, the odds ratios and confidence intervals were OR = 2.88 and 95% CI = 1.18–7.01 for governmental official websites and media (MHLW) and OR = 0.39 and 95% CI = 0.15–1.00 for word of mouth from family/friends.

## 4. Discussion

In this study, we conducted a KAP survey for pharmacists involved in community vaccine preparation. The majority of the pharmacists were shown to have positive KAP, and most of the pharmacists’ sources were news media (*n* = 124, 87.9%) and government office sites (*n* = 101, 71.6%). In addition, 8.5% of the pharmacists obtained information from academic journals, which was less than 10% of all participants. These results showed that conventional mass media and the internet are important information sources for pharmacists, and the current situation in which many pharmacists in Japan have not been able to obtain the latest information from academic journals was indicated. Knowledge was shown to be associated with attitudes and practices in previous KAP surveys related to COVID-19 [29,30,31], and in this study as well, there was a correlation among knowledge, attitudes, and practices. Our results suggested that each item of KAP was mutually influential on the others, and that all items were essential components.

Previous KAP surveys of COVID-19 infection control among Japanese pharmacists reported lower KAP scores [23]. It has also been reported that the KAP scores of pharmacists were lower than those of doctors and nurses abroad [32,33]. However, our study showed good scores for all items of KAP. Awareness and comprehension of infection control measures may have increased as pharmacists became practically involved in COVID-19 infection control measures through participating in vaccine preparation. As a result, knowledge scores may have increased. At the same time, the pharmacists’ KAP scores were lower than those previously reported for doctors and nurses, possibly because the pharmacists were not involved in COVID-19 infection control. Therefore, it was considered that practices were indispensable to obtain good knowledge scores.

The aggregated results for attitude scores revealed that 90% or more pharmacists answered positively for six attitude items. Of these, almost all pharmacists answered “I would like to be vaccinated with COVID-19” and “I think that vaccination can prevent the spread of COVID-19”. In addition, four items such as “I think all information related to COVID-19 needs to be shared with other healthcare professionals”, “I think pharmacists can contribute to vaccination of the local residents from the public health and social aspects”, “I think it is the pharmacist’s social mission to work together to curb the spread of COVID-19 infection”, and “I understand that this infection is very infections” had more than 90% of the pharmacists answering “I would think so”. Therefore, many pharmacists indicated positive attitude scores. Conversely, only 31.9% of the pharmacists answered “I would think so” to the question “Ultimately, we can completely control COVID-19 infection”, indicating the difficulty and seriousness of COVID-19 infection control.

As a result of extracting the related factors of KAP using stepwise logistic regression analysis, the related factors for knowledge and attitudes scores were extracted. Regarding knowledge scores, age was extracted as in previous studies [23], with the lowest knowledge scores for the respondents in their 20s. It may be conceivable that people in their 20s have a lower risk of becoming severely ill with COVID-19, and it is thought that the percentage of pharmacy managers who manage pharmacies is small. Conversely, regarding the attitude score, four items were extracted: “marriage”, “pharmacist careers”, “information source: official government website”, and “information source: word of mouth from family and friends”. For “marital status”, the attitude scores were significantly higher in unmarried than in married individuals. This is thought to be because the percentage of women pharmacists in Japan is higher than that of men. In other words, it is conceivable that married women take more time to do housework and raise children, and it is difficult for them to be actively involved in COVID-19 infection control as a pharmacist. In addition, pharmacists with less than 10 years of experience had the most positive attitude scores in pharmacist history, and pharmacists with more than 10 years and less than 20 years had significantly lower scores. This suggested that attitude scores fluctuated according to the length of the pharmacists’ career. Furthermore, the attitude scores of pharmacists who obtained information from the official website of the government were significantly high, and those of pharmacists who obtained information by word of mouth from family and friends were significantly low. These findings suggest that the presence or absence of active information collection is associated with attitude scores. As for the practice scores, as in the previous reports [19,34], no relevant vital factors were extracted, and a significant relationship was found between knowledge and attitudes. From this, we suggest that practical involvement with vaccine preparation brought about knowledge and attitude improvements.

In this study, gender was not significantly associated with KAP scores. However, a KAP survey conducted with the general public reported an association between gender and knowledge scores [35,36]. This may be due to women being more likely than men to feel anxious regarding a variety of issues [37], and because of their anxiety, they may become enthusiastic regarding collecting information on infection control. As a result, their knowledge scores may be higher. Conversely, there have been reports that no association exists between gender and knowledge in KAP surveys conducted with healthcare professionals [28,38,39]. It has been speculated that this is because healthcare workers are involved in COVID-19 infection control regardless of gender. On the other hand, a KAP survey administered to medical students reported an association between gender and knowledge scores [40]. This is probably because medical students are not yet directly involved in COVID-19 infection control as healthcare workers. Similarly, a KAP survey conducted with Japanese pharmacists during the period prior to vaccine preparation engagement, i.e., when they were not directly involved in COVID-19 infection control, showed a significant association between gender and knowledge scores [23]. From the above, we suggest that factors related to the KAP survey of COVID-19 were influenced by the practical environment of the study participants.

Regarding pharmacists’ sources of information, most sources were news and government office sites, and few pharmacists obtained information from academic journals. This is a similar result to a KAP survey conducted in Japan before vaccine development [23]. Recently, the use of SNS has also been considered a useful information resource [19,41]. However, news alone does not alter the amount of information available compared to the general public, and SNS alone may be affected by erroneous information. As medical professionals involved with vaccinations need to obtain general information from the news, they also need to obtain official information distributed by the government and the latest specialized information from academic journals.

There are several limitations to this study. First, the survey was limited to pharmacists in Tokyo, and no results were obtained from pharmacists in other cities. Second, because of the cross-sectional nature of the study, a causal relationship cannot be determined. Third, the questionnaires may not correspond exactly to what the health professional thinks and does in clinical practice. Fourth, the small sample size of the study may not reflect the opinions of the entire pharmacist community. Further, the response rate was 55.3%—a little low. It is possible that some pharmacists have stopped participating in web questionnaires because they are not familiar with the operation. However, in Japan, the same information is available from media news in all regions, and in fact, pharmacists are involved in the development and manufacture of vaccines. Therefore, it is unlikely that knowledge, behaviors, and attitudes regarding infection control will vary significantly from region to region and from individual to individual. The distribution of gender and age in this study was similar to the distribution of the whole pharmacist community in Japan. Therefore, it is considered that the bias of the obtained results is small.

The points that should be emphasized in this study are that this was a cross-sectional study conducted by a Japanese pharmacist during the period when he was engaged in vaccine preparation to prevent the spread of COVID-19 in the local population. The survey was conducted at a time when the number of people infected with COVID-19 per day in Tokyo was rapidly increasing and the crisis of medical collapse was being hailed. Local pharmacists were involved in vaccine preparation, even on holidays, so that as many citizens as possible could get their first vaccination as soon as possible. In addition, community pharmacist associations actively held training sessions on vaccine preparation for pharmacists. Previously, pharmacists in Japan only dispensed prescriptions based on individual pharmacies and were not openly involved in activities such as community infection control. In view of these circumstances, the results of this study were obtained during the first practical involvement of Japanese pharmacists in community infectious disease control in a critical setting of a spreading COVID-19 infection. This can be useful information when considering the public health significance and social value of community pharmacists who were practically participating in infectious disease countermeasures, such as vaccination, not only in Japan but also globally. Indeed, pharmacists’ knowledge and attitude scores under situations where practical involvement was sought were very positive. Furthermore, because more than 90% of the pharmacists replied that it is the duty of pharmacists to engage in public health, such as infection control measures, it can be said that pharmacists are very positive and highly aware of public health efforts. Moreover, as community pharmacists’ knowledge, attitude, and practice scores toward COVID-19 were positive, the active utilization of pharmacists could lead to better community public health and could provide great added value to patients and the entire healthcare system. Therefore, the use of community healthcare professionals, such as pharmacists, may expand the potential of human medical resources and may consequently lead to the activation of community healthcare. It is hoped that policymakers understand the value of pharmacists as human resources and effectively utilize them for the development of public health. Furthermore, we believe that the results of this study may contribute to strengthening the role of pharmacists in communities in different parts of the world, not only in Japan.

## 5. Conclusions

The KAP of pharmacists on COVID-19 was very positive under situations of practical engagement in vaccination. This suggests that utilizing pharmacists may increase the professionalism of pharmacists and may provide new added value to patients and the health care system as a whole. It is hoped that policymakers will understand the value of pharmacists, who are healthcare professionals, as human resources, and make effective use of them, thus contributing to the development of public health.

## Figures and Tables

**Figure 1 ijerph-19-09035-f001:**
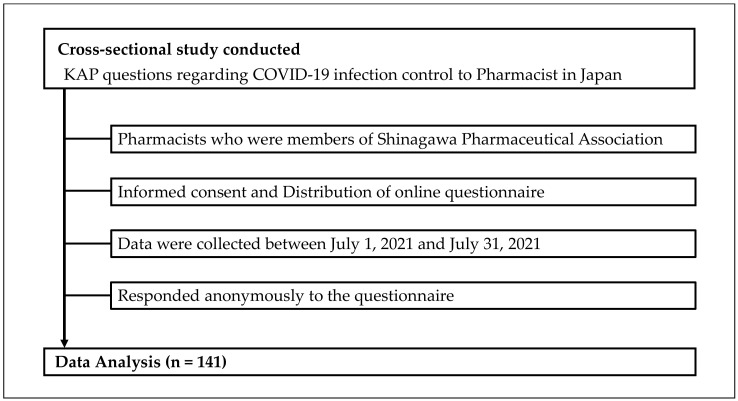
Flowchart of the cross-sectional study.

**Table 1 ijerph-19-09035-t001:** Participant characteristics (*n* = 141).

Variable	
**Age (years)**	44.8 ± 11.9
**Sex, *n* (%)**	
Men	49 (34.8%)
Women	92 (65.2%)
**Marital status, *n* (%)**	
Married	97 (68.8%)
Unmarried	44 (31.2%)
**Academic background, *n* (%)**	
4-year university	93 (66.0%)
6-year university	32 (22.7%)
Master’s degree/Doctoral degree	16 (11.3%)
**Pharmacist career, *n* (%)**	
Less than 10 years	46 (32.6%)
10 years or more and less than 20 years	39 (27.7%)
Over 20 years	56 (39.7%)
**Information source of COVID-19, *n* (%)**	
Official website and media of the WHO or CDC	32 (22.7%)
Official government website and media	101 (71.6%)
Media such as news (TV, radio, internet, magazines, and newspapers)	124 (87.9%)
Social media (Facebook, Twitter, and Instagram)	50 (35.5%)
Word of mouth from family/friends	28 (19.9%)
Original paper	12 (8.5%)
I don’t remember	2 (1.4%)

**Table 2 ijerph-19-09035-t002:** Respondents’ answers on COVID-19 knowledge items.

		Correct Answer	Incorrect Answer	Don’t Know
K1	COVID-19 is caused by beta coronavirus.	59 (41.8%)	16 (11.3%)	66 (46.8%)
K2	COVID-19 is transmitted by food intake.	100 (70.9%)	24 (17.0%)	17 (12.1%)
K3	Common clinical symptoms of COVID-19 are fever, dry cough, dyspnea, and malaise.	128 (90.8%)	8 (5.7%)	5 (3.5%)
K4	Sneezing, runny nose, stuffy nose, and headache are less common symptoms of COVID-19.	36 (25.5%)	96 (68.1%)	9 (6.4%)
K5	PCR can be used to diagnose COVID-19.	126 (89.4%)	11 (7.8%)	4 (2.8%)
K6	Washing your hands with soap and water for at least 30 s is effective in preventing COVID-19 infection.	132 (93.6%)	6 (4.3%)	3 (2.1%)
K7	Loss of taste and smell is characteristic of COVID 19 infection.	129 (91.5%)	9 (6.4%)	3 (2.1%)
K8	Symptom-free COVID-19 patients (during the incubation period) do not spread the virus to others.	136 (96.5%)	0 (0%)	5 (3.5%)
K9	COVID-19 infection spreads through the infected person’s respiratory droplets.	137 (97.2%)	1 (0.7%)	3 (2.1%)
K10	The incubation period of coronavirus is 1 to 14 days.	126 (89.4%)	8 (5.7%)	7 (5.0%)
K11	Elderly people, patients with chronic illness, DM, COPD can be severe.	137 (97.2%)	0 (0%)	4 (2.8%)
K12	Shaking hands and avoiding crowded areas and public transport can prevent COVID-19 infection.	135 (95.7%)	2 (1.4%)	4 (2.8%)
K13	Keeping social distance is effective in preventing the spread of COVID-19.	135 (95.7%)	1 (0.7%)	5 (3.5%)
K14	Antibiotics are the first-line drug if COVID-19 infection is suspected or confirmed.	121 (85.8%)	8 (5.7%)	12 (8.5%)
K15	Early response and supportive care are effective in recovering from infection, as there is no effective treatment for COVID-19.	123 (87.2%)	8 (5.7%)	10 (7.1%)
K16	Isolating and treating individuals infected with COVID-19 is an effective way to break the chain of infection.	138 (97.9%)	0 (0%)	3 (2.1%)
K17	Large-scale group activities can spread COVID-19 infection.	138 (97.9%)	0 (0%)	3 (2.1%)
Knowledge Score		14.4 ± 2.5		

Of the questions K1–K17, “No” is the correct answer for questions K2: “COVID-19 is transmitted by food intake”, K8: “Symptom-free COVID-19 patients (during the incubation period) do not spread the virus to others”, and K14: “Antibiotics are the first-line drug if COVID-19 infection is suspected or confirmed”. “Yes” is the correct answer for all the other questions—K1, K3, K4, K5, K6, K7, K9, K10, K11, K12, and K13.

**Table 3 ijerph-19-09035-t003:** Respondents’ answers on COVID-19 attitudes.

		Yes	No
A1	I would like to be vaccinated with COVID-19 (answer “yes” if already vaccinated).	133 (94.3%)	8 (5.7%)
A2	I am motivated to be involved in citizens’ vaccination work (such as vaccine filling).	112 (79.4%)	29 (20.6%)
A3	I don’t think there is any problem even if vaccination by a pharmacist is approved as in overseas.	96 (68.1%)	45 (31.9%)
A4	If vaccination by a pharmacist is approved, I am willing to vaccinate the public.	90 (63.8%)	51 (36.2%)
A5	If vaccination by a pharmacist is approved at a dispensing pharmacy like overseas, I think it will be a great advantage for local residents.	115 (81.6%)	26 (18.4%)
A6	If vaccination becomes possible at local pharmacies, I think it can contribute to improving the vaccination rate of local residents.	116 (82.3%)	25 (17.7%)
A7	I think pharmacists can contribute to vaccination of local residents from the public health and social aspects.	129 (91.5%)	12 (8.5%)
A8	In order to popularize antibody testing, I think it’s okay for pharmacists to do antibody testing at pharmacies.	109 (77.3%)	32 (22.7%)
A9	I think that vaccination can prevent the spread of COVID-19.	133 (94.3%)	8 (5.7%)
A10	Ultimately, I think we can completely control the infection of COVID-19.	45 (31.9%)	96 (68.1%)
A11	I think healthcare professionals need to be aware of all information regarding COVID-19.	122 (86.5%)	19 (13.5%)
A12	I think all information related to COVID-19 needs to be shared with other healthcare professionals.	136 (96.5%)	5 (3.5%)
A13	I think the spread of COVID-19 can be prevented by taking the precautionary measures indicated by WHO and the government.	97 (68.8%)	44 (31.2%)
A14	I think it is necessary to use gowns, gloves, masks, and face shields when dealing with patients with COVID-19.	117 (83.0%)	24 (17.0%)
A15	I think it is the pharmacist’s social mission to work together to curb the spread of COVID-19 infection.	138 (97.9%)	3 (2.1%)
A16	I understand that this infection is very infectious.	138 (97.9%)	3 (2.1%)
Attitude score		13.0 ± 2.6	

**Table 4 ijerph-19-09035-t004:** Respondents’ answers on COVID-19 practices.

		Yes	No
P1	I have attended workshop for COVID-19 infection prevention.	76 (53.9%)	65 (46.1%)
P2	I keep a social distance with others to prevent infection.	138 (97.9%)	3 (2.1%)
P3	When I touch the front door, which is touched by an unspecified number of people, I open and close it while being careful of infection.	120 (85.1%)	21 (14.9%)
P4	I wash my hands with soap routinely.	138 (97.9%)	3 (2.1%)
P5	I avoid crowds as much as possible.	129 (91.5%)	12 (8.5%)
P6	I avoid meeting friends and relatives.	125 (88.7%)	16 (11.3%)
P7	I try not to touch my eyes, nose, and mouth as much as possible.	124 (87.9%)	17 (12.1%)
P8	When dealing with the belongings of a patient suspected of having a corona infection, I take care to prevent infection.	136 (96.5%)	5 (3.5%)
P9	I try not to take the elevator as much as possible.	58 (41.1%)	83 (58.9%)
P10	When I sneeze or cough, I cover my nose and mouth with a tissue or handkerchief.	137 (97.2%)	4 (2.8%)
P11	To prevent infection, I always try to throw used tissue in the trash.	138 (97.9%)	3 (2.1%)
Practice score		9.4 ± 1.4	

**Table 5 ijerph-19-09035-t005:** Correlations between knowledge, attitude, and practice scores.

	Correlation (r)	*p* Value
Knowledge and Attitude	0.44	<0.001
Knowledge and Practice	0.21	0.01
Attitude and Practice	0.32	<0.001

**Table 6 ijerph-19-09035-t006:** Evaluation of attribute factors related to KAP using univariate logistic regression analysis.

Characteristics	Category	Good Knowledge			Good Attitude			Good Practice		
		OR	95% CI	*p* Value	OR	95% CI	*p* Value	OR	95% CI	*p* Value
Age	20–29 years	1			1			1		
	30–39 years	6.3	1.58–25.08	0.009	1.11	0.34–3.64	0.859	2.49	0.64–9.70	0.188
	40–49 years	11.1	2.52–48.84	0.001	1.75	0.54–5.67	0.351	1.85	0.53–6.38	0.332
	50–59 years	3.75	1.06–13.31	0.041	1.06	0.32–3.48	0.923	1.33	0.38–4.67	0.659
	60 over	16.2	1.78–147.06	0.013	1.00	0.27–3.74	1.000	2.46	0.51–11.80	0.260
Sex	Men	0.52	0.21–1.28	0.154	1.00	0.48–2.09	0.996	0.53	0.23–1.20	0.126
	Women	1			1			1		
Marital Status	Married	2.93	1.18–7.31	0.021	0.42	0.18–0.97	0.041	1.97	0.86–4.54	0.110
	Unmarried	1			1			1		
Academic background	4-year university	1			1			1		
	6-year university	0.30	0.11–0.78	0.014	0.70	0.30–1.59	0.391	1.04	0.40–2.74	0.934
	Master’s degree/Doctoral degree	0.94	0.19–4.70	0.939	2.06	0.55–7.79	0.285	2.04	0.43–9.71	0.370
Pharmacist career	Less than 10 years	1			1			1		
	10 years or more and less than 20 years	1.22	0.43–3.41	0.708	2.27	0.85–6.06	0.101	1.73	0.57–5.21	0.331
	Over 20 years	4.09	1.20–13.87	0.024	0.98	0.44–2.19	0.955	1.04	0.41–2.60	0.934
Source of information on COVID-19	Official website and media of the WHO or CDC	1.48	0.46–4.71	0.509	1.99	0.79–5.02	0.145	2.19	0.70–6.83	0.176
	Official government website and media	1.8	0.71–4.58	0.215	2.48	1.16–5.31	0.019	1.35	0.57–3.21	0.498
	Media such as news (TV, radio, internet, magazines, and newspapers)	1.11	0.29–4.24	0.874	0.4	0.11–1.48	0.172	0.77	0.21–2.88	0.697
	Social media (Facebook, Twitter, and Instagram)	1.04	0.41–2.65	0.941	1.05	0.50–2.19	0.907	1.13	0.48–2.64	0.784
	Word of mouth from family/friends	0.65	0.23–1.84	0.416	0.4	0.17–0.92	0.032	0.48	0.19–1.21	0.121
	Original paper	0.35	0.09–1.26	0.108	0.97	0.28–3.39	0.956	1.39	0.29–6.70	0.684

**Table 7 ijerph-19-09035-t007:** Extraction of attribute factors related to knowledge, attitudes, and practices by stepwise logistic regression analysis.

Characteristics	Category	Good Knowledge			Good Attitude		
		Adjusted OR	95% CI	*p* Value	Adjusted OR	95% CI	*p* Value
Age	20–29 years	1					
	30–39 years	6.3	1.58–25.08	0.009			
	40–49 years	11.1	2.52–48.84	0.001			
	50–59 years	3.75	1.06–13.31	0.041			
	60 over	16.2	1.78–147.06	0.013			
Sex	Men						
	Women						
Marital Status	Married				0.37	0.15–0.90	0.029
	Unmarried				1		
Academic background	4-year university						
	6-year university						
	Master’s degree/Doctoral degree						
Pharmacist career	Less than 10 years						
	10 years or more and less than 20 years						
	Over 20 years						
Source of information on COVID-19	Official website and media of the WHO or CDC						
	Official government website and media				2.81	1.26–6.26	0.012
	Media such as news (TV, radio, internet, magazines, and newspapers)						
	Social media (Facebook, Twitter, and Instagram)						
	Word of mouth from family/friends				0.38	0.16–0.91	0.030
	Original paper						

Stepwise logistic regression analysis was performed with Age, Sex, Marital Status, Academic Background, Pharmacist Career, and Source of Information on COVID-19 as explanatory variables. As a result, Age was extracted for Good Knowledge, and Marital Status, Official Government Website and Media, and Word of Mouth from Family/Friends were extracted as important variables for Good Attitude. Variables in blank cells were not extracted as significant variables.

## Data Availability

The data are not publicly available as all participants have not consented to the public disclosure of the data online. However, the data presented in this study are available on request from the corresponding author.

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
