# Peer review of "Survey of Pharmacists’ Knowledge, Attitudes, and Practices (KAP) concerning COVID-19 Infection Control after Being Involved in Vaccine Preparation: A Cross-Sectional Study"

_ijerph, 2022, doi:10.3390/ijerph19159035_

Round 1

Reviewer 1 Report

The authors conducted a cross-sectional study among pharmacists using a Google online questionnaire. The study shows Good KAP scores were recorded among the pharmacists. I have some concerns about the manuscript.

1. Table 1 pharmacist career, n(%) 16.1±10.5. It made me confused, the unit of the 16.1±10.5 is year or person?

2. Table 7 there are many blank cells in this table, especially academic background and Career. Why? What was the results for these cells? Is it necessary to put them in?

3. The response rate is a little low, while the authors did not mention in the discussion and limitation.

4. The results are presented clearly, while the conclusion can be improved. For instance, in the discussion section, the conclusions ’The KAP of pharmacists on COVID-19 may have greatly improved after their practical engagement in vaccination’ were not supported by the results. As this is a cross sectional study, there is no history results to be compared, it’s difficult to draw this conclusion.

Author Response

Dear Reviewer1

We wish to express our appreciation to the reviewers for their comments on our paper. The comments have helped us significantly improve the paper. 

The corrected parts of the manuscript are highlighted in yellow. Please check the revised manuscript.

Sincerely, 

Nobuyuki Wakui(Corresponding Author)

Reviewer 2 Report

In the study of Wakui et al. entitled “Survey of pharmacists’ Knowledge, Attitudes, and Practices (KAP) concerning COVID-19 infection control after being involved in vaccine preparation: A cross-sectional study”, the authors conducted a cross-sectional study among 141 pharmacists who were members of a pharmacist association in the Shinagawa Ward of Tokyo (July 1−31, 2021) using a Google forms platform. The survey included demographic information and KAP questions regarding COVID-19. A correlation test was used for analyzing KAP scores. Significant correlations were found between all KAP scores. Stepwise logistic regression analysis showed “age” as a significant knowledge factor and “marriage,” “pharmacist careers,” “information source: official government website,” and “information source: word of mouth from family and friends” as significant attitudinal factors. Good KAP scores were recorded in this study, indicating increased comprehension of infection control measures and increased knowledge scores as pharmacy, pharmacists were practically involved in COVID-19 infection control measures through vaccine preparation. Policymakers should understand the value of pharmacists as health care professionals and should enhance public health through the effective use of pharmacists. The study is important because investigates the perception and knowledge of health professionals about practices related to infection control during the COVID-19 pandemic. Thus, should be accepted after minor revisions.

Minor revisions

1) The new coronavirus is not COVID19, which means “coronavirus disease”. So, change to severe acute respiratory syndrome coronavirus 2 (SARS-CoV-2).

2) Likewise, the virus (SARS-CoV-2) is spread, not the disease. Please change too.

3) Regarding question K2 "COVID-19 is transmitted by food intake" the WHO says that "There is currently no evidence that people can catch COVID-19 from food, including fruits and vegetables. Fresh fruits and vegetables are part of a healthy diet and their consumption should be encouraged." Thus, authors should review the interpretation of this question and the answers related to it.

4) The same for the question K14 "Antibiotics are the first-line drug if COVID-19 infection is suspected or confirmed." The WHO says that "Antibiotics work only against bacteria, not viruses. COVID-19 is caused by a virus, and therefore antibiotics should not be used for prevention or treatment. Some people who become ill with COVID-19 can also develop a bacterial infection as a complication. In this case, antibiotics may be recommended by a health care provider." Thus, authors should review the interpretation of this question and the answers related to it. Furthermore, this issue increases the misperception that antibiotics should be used to treat viral infections.

5) Another limitation of the study is that questionnaires can always be answered in an unpretentious way, not corresponding exactly to what the health professional thinks and does in clinical practice.

Author Response

Dear Reviewer2

We wish to express our appreciation to the reviewers for their comments on our paper. The comments have helped us significantly improve the paper.

The corrected parts of the manuscript are highlighted in yellow. Please check the revised manuscript.

Sincerely,

Nobuyuki Wakui(Corresponding Author)
